# Irradiation Mediates *IFNα* and *CXCL9* Expression in Non-Small Cell Lung Cancer to Stimulate CD8^+^ T Cells Activity and Migration toward Tumors

**DOI:** 10.3390/biomedicines9101349

**Published:** 2021-09-29

**Authors:** Chun-Chia Cheng, Yi-Fang Chang, Ai-Sheng Ho, Zong-Lin Sie, Jung-Shan Chang, Cheng-Liang Peng, Chun-Chao Chang

**Affiliations:** 1Radiation Biology Research Center, Institute for Radiological Research, Chang Gung University/Chang Gung Memorial Hospital, Taoyuan 333, Taiwan; cccheng.biocompare@gmail.com; 2Division of Hematology and Oncology, Department of Internal Medicine, Mackay Memorial Hospital, Taipei 104, Taiwan; changyifang@gmail.com; 3Laboratory of Good Clinical Research Center, Department of Medical Research, Mackay Memorial Hospital, New Taipei City 251, Taiwan; 4Department of Medicine, Mackay Medical College, New Taipei City 252, Taiwan; 5Division of Gastroenterology, Cheng Hsin General Hospital, Taipei 112, Taiwan; aisheng49@gmail.com (A.-S.H.); zonlins@gmail.com (Z.-L.S.); 6Graduate Institute of Medical Sciences, School of Medicine, College of Medicine, Taipei Medical University, Taipei 110, Taiwan; js.chang@tmu.edu.tw; 7Institute of Nuclear Energy Research, Atomic Energy Council, Taoyuan 325, Taiwan; clpeng@iner.gov.tw; 8Division of Gastroenterology and Hepatology, Department of Internal Medicine, Taipei Medical University Hospital, Taipei 110, Taiwan; 9Division of Gastroenterology and Hepatology, Department of Internal Medicine, School of Medicine, College of Medicine, Taipei Medical University, Taipei 110, Taiwan

**Keywords:** non-small-cell lung cancer, *IFNα*, *CXCL9*, CXCR3, PD-1, CD8^+^ T cells, irradiation, radiotherapy

## Abstract

Irradiation-broken DNA fragments increase type I interferon and chemokines secretion in tumor cells. Since radiotherapy may augment tumor immunotherapy, we hypothesize that the chemokines increased by irradiation could recruit CD8^+^ T cells to suppress tumor proliferation. This study intended to unveil the secreted factors activating and recruiting CD8^+^ T cells in non-small-cell lung cancer (NSCLC). EGFR-positive A549 was selected and treated by X-irradiation (IR) to identify the overexpression of chemokines associated to CD8^+^ T cell cytotoxicity and recruitment. A transwell assay with Alexa 488-labeled CD8^+^ T cells was used to evaluate CD8^+^ T cell motility in vitro. A nuclear imaging platform by In^111^-labeled nivolumab was used to track CD8^+^ T cells homing to tumors in vivo. The activation markers *GZMB*, *PRF-1*, and *IFNγ*, migration marker *CD183* (CXCR3), and inhibitory marker *CD274* (PD-1), were measured and compared in CD8^+^ T cells with A549 co-cultured, chemokines treated, and patients with late-stage lung cancer. We found that IR not only suppressed A549 proliferation but also induced *IFNα* and *CXCL9* expression (*p* < 0.05). *IFNα* majorly increased *IFNγ* levels in CD8^+^ T cells (*p* < 0.05) and synergistically with *CXCL9* enhanced CD8^+^ T cell migration in vitro (*p* < 0.05). We found that CXCR3 and PD-1 were down-regulated and up-regulated, respectively, in the peripheral blood CD8^+^ T cells in patients with lung cancer (n = 4 vs. healthy n = 3, both *p* < 0.05), which exhibited reduction of cell motility (*p* < 0.05). The in vivo nuclear imaging data indicated highly CD8^+^ T cells migrated to A549-induced tumors. In addition, we demonstrated that healthy PBMCs significantly suppressed the parallel tumor growth (*p* < 0.05) and the radioresistant tumor growth in the tumor xenograft mice (*p* < 0.05), but PBMCs from patients with lung cancer had lost the anti-tumor capacity. We demonstrated that IR induced *IFNα* and *CXCL9* expression in A549 cells, leading to CD8^+^ T cell migration. This study unveiled a potential mechanism for radiotherapy to activate and recruit CD8^+^ T cells to suppress lung tumors.

## 1. Introduction

Lung cancer is the most common type and the leading cause of cancer-related deaths worldwide, including small cell lung cancer and non-small cell lung cancer. The most common type is non-small cell lung cancer (NSCLC) which comprises 80% of lung cancers. In this disease, squamous cancer and small cell lung cancer are caused mostly by smoking; but non-small cell lung cancer, such as adenocarcinoma, mostly appears in non-smokers [1]. In Asia, most patients have EGFR-positive lung cancer with Exon19 deletion and Exon21 L858R mutation [2], which is suitable for targeted therapy such as tyrosine kinase inhibitors in clinical practice [3,4].

There are several therapeutics against lung cancer, including surgery, radiation therapy, chemotherapies, targeted therapies, immunotherapies, and cell therapies. Immunotherapies are particularly promising in current clinical practice, resulting in very good therapeutic outcomes [5]. Although immunotherapy is reliable, the selection of treatment for lung cancer is mainly dependent on the stage, the characteristic types, and the patient’s immune diversity. Inevitably, basic treatments such as radiotherapy are often preferred for patients with early-stages of lung cancer or are used combined with other therapies. According to statistics, radiotherapy is used at least once in over 60% of lung cancer patients until the cure of this disease [6]. CD8^+^ T cells are the main target of clinical immunotherapies such as blocking immune checkpoints by targeting PD-1 and CTLA4 augments the CD8^+^ T cell activity and cytotoxicity [7]. This study aims to investigate the radiotherapy effect on lung cancer cells and investigate whether radiotherapy increases immune system activity, particularly for CD8^+^ T cells since CD8^+^ T cells are the main eradicator of tumors.

In solid cancers, tumors can be classified as hot, cold, altered-immunosuppressed and altered-excluded tumors, based on the degree of tumor CD8^+^ T-cell infiltration in the tumor microenvironment [8]. Hot tumors are characterized by high infiltration of CD8^+^ T cells and respond with better immunotherapeutic outcomes, whereas cold tumors are not. The treatment of cold tumors with Vps34 inhibitors induces the release of inflammatory chemokines such as *CCL5* and *CXCL10* in tumor cells, resulting in homing of natural killer (NK) and CD8^+^ T cells to the tumor microenvironment [9]. Irradiation (IR) causes tumor DNA damage and results in consequent tumor apoptosis [10]. Previous studies have demonstrated that low doses of fractionated IR significantly improve CD8^+^ T cell-mediated tumor remission in combination with anti-PD-1 or anti-PD-L1 therapies [11,12,13]. In IR treatment, the accumulation of cytosolic DNA fragments activates the cGAS-STING signaling pathway [14], inducing type I interferon (IFN) secretion to potentially reactivate immune surveillance against tumors. Moreover, IR enhances the secretion of cytokines [15], resulting in an increase of the homing rate of immune cells to the tumor microenvironment [16,17]. Therefore, IR-mediated tumor therapies are considered for improving the anti-tumor efficacy of clinical immunotherapies [18,19,20].

To investigate the mechanism of radiotherapy reactivating CD8^+^ T cells, we investigate the type 1 IFNs and chemokines expression in the irradiated lung cancer A549 cells. The in vitro CD8^+^ T cell migration assay and in vivo nuclear imaging platform were used to determine the CD8^+^ T cell migratory capacity. Meanwhile, characteristics of CD8^+^ T cells were investigated between healthy volunteers and patients with lung cancer, including the activation and inhibitory markers expression, cell motility, and anti-tumor capacity in this study.

## 2. Methods

### 2.1. Healthy Volunteers and Tumor Patients

Blood samples were acquired from healthy volunteers and patients with lung cancer at Mackay Memorial Hospital, Taipei, Taiwan (Table 1). The study protocol was approved by the regulatory authorities and Institutional Review Boards at Mackay Memorial Hospital (20MMHIS018e, 7 May 2020). Signed and informed written consent were obtained from all participants, and all research was performed in accordance with the relevant guidelines and regulations.

### 2.2. Cell Culture

All the lung cancer cell lines used in this study were purchased from the American Type Culture Collection (ATCC, Manassas, VA, USA) and were reauthenticated through short tandem repeat profiling (Applied Biosystems, Waltham, MA, USA). A549 cells were cultured in Dulbecco’s Modified Eagle’s Medium (DMEM). PC9 and HCC827 were cultured in Roswell Park Memorial Institute (RPMI) 1640 medium. The tumor cells cultured in the medium were supplied with 10% fetal bovine serum (FBS) and 1% penicillin-streptomycin (P/S). All cells were cultured at 37 °C with 5% CO_2_. 

### 2.3. Cell Proliferation

The WST-1 assay (Sigma-Aldrich, Burlington, MA, USA) was used to measure the cell viability according to the manufacturer’s protocol. For investigating the anti-proliferation effect of irradiation, A549 cells were exposed to 20 Gy of X-ray and cell viability was detected post 24 h- and 48 h-cultured, and 1 × 10^3^ A549 cells were seeded in a 96-well plate and a 4-times repeat was used.

### 2.4. Colony Formation

A colony formation assay was used to evaluate the anti-tumorigenicity effect of irradiation. A549 cells were exposed to 20 Gy of X-ray, and 1 × 10^3^ cells A549 cells were seeded in a 6-well plate and 3-times repeat was used. Colony formation was observed under a microscope post 14 days cultured at 37 °C with 5% CO_2_, and the cell numbers per view were measured by ImageJ (National Institutes of Health, Bethesda, Maryland, USA, https://imagej.nih.gov/ij/ accessed on 5 July 2021).

### 2.5. Isolation of Peripheral Blood Mononuclear Cells (PBMCs) and CD8^+^ T Cells

The procedure for PBMCs and CD8^+^ T cell isolation was the same as described previously [21]. The detailed protocol is shown in Appendix A.

### 2.6. Flow Cytometry

The 2 × 10^5^ tumor cells or PBMCs were resuspended in 100 μL of culture medium and were incubated with fluorescent reagents for 30 min at room temperature. To detect T cell markers in PBMCs, cells were incubated with fluorescent-conjugated antibodies including APC/Cy7-CD3, PE/Cy7-CD4, Alexa488-CD8, PE-Cy5.5-CD8, PE-CD16, PerCP/Cy5.5-CD25 (IL2Rα), PerCP/Cy5.5-CD28, Pacific blue-CD45, PerCP-Cy5.5-CD45RA, APC-*CD122* (IL2Rβ), PE-CD178 (FasL), Alexa700-CD183 (CXCR3), FITC-CD197 (CCR7) and APC-*CD279* (PD-1) (BioLegend, San Diego, CA, USA). To detect cell apoptosis in A549 cells treated with 20 Gy of irradiation, cells were incubated with Annexin V-FITC and Propidium Iodide (Strong Biotech Corporation, Taiwan). Cells were consequently added to 900 μL of PBS buffer containing 0.1% of FBS and analyzed using an Attune NxT Flow Cytometer (Invitrogen, Waltham, MA, USA).

### 2.7. Quantitative Polymerase Chain Reaction (qPCR)

The procedures for mRNA extraction and complementary DNA preparation were the same as those described previously [22]. Quantitative polymerase chain reaction (qPCR) was performed using the SYBR Green system (Applied Biosystems, Foster City, CA, USA) according to the manufacturer’s instructions. The quantification was shown based on 3-times repeat. The primers are shown in Table 2.

### 2.8. In Vitro CD8^+^ T Cell Migration Assay

A transwell migration assay (3.0 μm) was used to detect CD8^+^ T cell migration capacity (JET biofil, Guangzhou, China). The detailed protocol is shown in Appendix A. The quantification was calculated based on 3-times repeat.

### 2.9. Animal

Male ASID mice (*Cg-Prkdcscid Il2rgtm1Wjl/YckNarl*) were purchased from the National Laboratory Animal Center, Taiwan. The 5-week-old mice were housed in a 12 h-light cycle at 22 °C. The animal studies were approved by the institutive ethical review committee in Chang Gung University, Taiwan. The tumor xenografts were established by injecting 2 × 10^6^ of A549 or HCC827 cells into the subcutaneous legs with or without 5 × 10^6^ PBMCs co-injected with A549 or PC9. Tumor volumes were recorded and calculated using the formula: 0.52 × width^2^ × length, whereas the width represents the smaller tumor diameter.

### 2.10. In Vivo PD-1 Nuclear Imaging Assay

In vivo PD-1 nuclear imaging was used to track the homing capacity of CD8^+^ T cells in the tumor xenograft mice (n = 3) based on ^111^In-labeled Nivolumab. The detailed protocol is shown in Appendix A.

### 2.11. Enzyme-Linked Immunosorbent Assay (ELISA) for Measurement of CXCL9

The Human *CXCL9* PicoKine ELISA (Boster, Pleasanton, CA, USA) was used according to the manufacturer’s protocol to determine the *CXCL9* concentration in the medium of A549 cells treated by 20 Gy irradiation and in the serum from the healthy volunteers (n = 3) and patients with lung cancer (n = 5). The detailed protocol is shown in Appendix A.

### 2.12. Statistical Analysis

GraphPad Prism V8.01 (GraphPad Software, Inc., San Diego, CA, USA) was used to analyze the statistically significant differences, whereas the unpaired two-tailed Student’s *t*-test was used to compare every two groups. Pearson’s correlation was used to calculate the correlation coefficient between the markers of CD8^+^ T cells in healthy volunteers and patients with lung cancer, in which r = −0.3~0.3: poor correlation; r: 0.3~0.6 and −0.3~−0.6: medium correlation; r = 0.6~0.9 and −0.6~−0.9: high correlation; r = 1 and−1: complete correlation. Moreover, *p* < 0.05 was considered to indicate a statistically significant difference.

## 3. Results

### 3.1. Irradiation Suppresses Tumor Growth and Induces IFNα, ISG15, and CXCL9 Expression in A549 Cells

We validated that 20 Gy of irradiation significantly suppressed A549 tumor proliferation (*p* < 0.001, Figure 1A) and colony formation (*p* < 0.001, Figure 1B). Meanwhile, irradiation increased cell apoptosis detected using a flow cytometer targeting Annexin V (Figure 1C). To investigate whether radiotherapy activated CD8^+^ T cells, type I interferon α and β subtype, irradiation marker ISG15, and the chemokines associated with CD8^+^ T cell migration were investigated. We found that irradiation increased *IFNα* and *ISG15* mRNA levels in irradiated A549 cells (*p* < 0.05, Figure 1D), it was potentially induced by activation of the cGAS-STING signaling pathway [14]. Moreover, the chemokines, including *CCL2*, *CCL3*, *CCL4*, *CCL5*, *CCL20*, *CXCL2*, *CXCL3*, *CXCL9*, *CXCL10*, *CXCL11*, *CXCL16*, and *CX3CL1* were measured in A549 treated with irradiation post 24 h. We found that irradiation significantly increased *CCL2*, *CCL4*, *CXCL2*, *CXCL3*, *CXCL9*, *CXCL10*, and *CX3CL1* (*p* < 0.05, Figure 1E). In addition, *CXCL9* was increased in the medium of irradiated A549 cells compared to parental A549 cells (*p* < 0.05, Figure 1E). The irradiation-treated A549 medium was collected and incubated with healthy PBMCs for 24 h. The CD8^+^ T cells were consequently isolated and investigated. We demonstrated that irradiation-treated A549 medium significantly increased *IFNγ* and *CD122* (*p* < 0.05, Figure 1F), whereas *IFNγ* was an activation marker of CD8^+^ T cells. *IFNα*, ISG15, and *CXCL9* were selected for further investigation for their effects on CD8^+^ T cells. CD8^+^ T cells were treated with the individual 20 ng/mL of *IFNα*, ISG15, and *CXCL9* for 2 h, and *GZMB*, *PRF1*, *IFNγ*, and *CD279* (PD-1) were investigated and compared. It demonstrated that only *IFNα* significantly increased *IFNγ* (*p* < 0.05), and slightly increased cytotoxic *GZMB* and *PRF1* expression and immune checkpoint *CD279* (PD-1) expression in CD8^+^ T cells (Figure 1G).

### 3.2. Irradiation-Treated A549 Medium Increased CD8^+^ T Cell Migration In Vitro

Since the *CXCL9* was demonstrated to increase in the irradiation-treated A549 cells, we intended to investigate whether the irradiation-treated A549 medium increased CD8^+^ T cell migration in vitro, where a 0.3 μm transwell plate was used to detect the migratory ability of the Alexa488-labeled CD8^+^ T cells (Figure 2A). We found that A549 co-cultured with nonCD8^+^ PBMCs were able to increase CD8^+^ T cell migration (*p* < 0.05, Figure 2A,B). In addition, irradiation-treated A549 medium significantly enhanced CD8^+^ T cell migration in a dose-dependent manner (*p* < 0.05, Figure 2A,B). To clarify the effect of nonCD8^+^ PBMCs on CD8^+^ T cell migration, the supernatant of A549 treated with nonCD8^+^ PBMCs for 24 h was collected and loaded on a down well to attract CD8^+^ T cell migration (Figure 2C). The results demonstrated that A549 with nonCD8^+^T PBMCs supernatant significantly enhanced CD8^+^ T cell migration (*p* < 0.001, Figure 2C,D). We consequently demonstrated that *CCL2*, *CCL3*, *CCL4*, *CXCL2*, *CXCL3*, *CXCL9* and *CXCL10* increased in the nonCD8^+^ PBMCs after incubating with A549 for 24 h (*p* < 0.05, Figure 2E). Moreover, we found that *IFNα* and *CXCL9* increased CD8^+^ T cells migration in vitro (*p* < 0.05, Figure 2F,G).

### 3.3. A549 Cells Stimulated CD8^+^ T Cells Expressing IL2 to Induce GZMB and PRF1 Expression

We have previously demonstrated that healthy CD8^+^ T cells expressed highly granzyme B in co-cultured with lung cancer HCC827 and A549 cells in vitro [21]. We intended to investigate whether irradiated A549 also activated CD8^+^ T cells, the healthy CD8^+^ T cells were isolated and co-cultured with parental A549 and irradiated A549 cells and *GZMB*, *PRF1*, *IFNγ*, *CD122* (IL2Rβ), *CD279* (PD-1), and other markers of CD8^+^ T cells were measured. We found that irradiated lung cancer A549 cells were induced dominantly *IFNγ* in CD8^+^ T cells, higher than in parental A549 (*p* < 0.05, Figure 3A). However, parental A549 induced higher *GZMB* and *PRF1* in CD8^+^ T cells than in irradiated A549 (both *p* < 0.05, Figure 3A). The difference is based on CD8^+^ T cells being able to produce IL2 when co-cultured with A549 cells (*p* < 0.05, Figure 3B) but irradiation induced *IFNα* (Figure 1D) and consequently *IFNα* induced *IFNγ* expression (Figure 1G). IL2 significantly induced *GZMB* and *PRF1* in CD8^+^ T cells (both *p* < 0.05, Figure 3C).

### 3.4. CXCR3 Decreased in CD8^+^ T Cells of Patients with Lung Cancer That Lost In Vitro Migratory Capacity

*CXCL9* inducing CD8^+^ T cell motility is dependent on CXCR3 [23,24]. At first, the percentage of CD4^+^ and CD8^+^ T cells were isolated and compared between healthy volunteers (n = 4, N1 to N4 in Table 1) and patients with lung cancer (n = 4, P1 to P4 in Table 1). There was no significant difference in percentage and ratio of CD4^+^ to CD8^+^ T cells (Figure 4A). To investigate and compare the significant CD8^+^ T cell markers between healthy volunteers and patients with lung cancer, a flow cytometer was used for further detection of CD16, CD25 (IL2Rα), CD28, *CD122* (IL2Rβ), CD178 (FasL), CD183 (CXCR3), and *CD279* (PD-1) (Appendix A). We found that *CD122* and *CD279* increased and CD28 and CD183 decreased in the patients with lung cancer (n = 4) compared to that in healthy volunteers (n = 3, whereas N2 was excluded sine PD1 highly expressed, Appendix A) (all *p* < 0.05, Figure 4B). Moreover, *CD122* was positively associated with *CD279* (Figure 4C, R^2^ = 0.997, *p* < 0.001) and CD183 was negatively associated with *CD122* (R^2^ = 0.567, *p* = 0.045) and *CD279* (R^2^ = 0.553, *p* = 0.055) (Figure 4C). To investigate CXCR3 expression in different kinds of CD8^+^ T cell subtype, a flow cytometer was used to detect naïve, central memory (CM), effector memory (EM), and terminally differential (TD) CD8^+^ T cells based on selection by CCR7 and CD45RA (Figure 4D). The results revealed less naïve (19.6%) and EM (29.4%) in a patient with lung cancer compared to a healthy volunteer with 29.9% naïve and 56.4% EM (Figure 4D). Meanwhile, there was a higher TD (50.6%) in a patient than 11.9% TD in a healthy volunteer (Figure 4D). The CXCR3 expressional ratio was similar to that in subtypes of CD8^+^ T cells between a healthy volunteer and a patient with lung cancer (Figure 4E). Moreover, we validated that the CD8^+^ T cell migratory capacity was positively associated with CXCR3 expression in the enrolled samples (Figure 4F, R^2^ = 0.183, *p* = 0.338). To investigate the patient’s CD8^+^ T cell cytotoxicity, co-cultured with A549 cell in vitro, *GZMB* and *PRF1* as the cytotoxic markers were measured and compared. We demonstrated that the patient’s CD8^+^ T cells still expressed *GZMB* (*p* < 0.001) and *PRF1* (*p* < 0.05), but *PRF1* was lower than that in healthy CD8^+^ T cells (*p* < 0.05, Figure 4G). In addition, we found lower cell motility in the CD8^+^ T cells in the patients with lung cancer compared to in healthy CD8^+^ T cells (*p* < 0.05, Figure 4H), and there were decreased levels of *CXCL9* and *CXCL10* in the patient’s nonCD8^+^ PBMCs compared to healthy nonCD8^+^ PBMCs (*p* < 0.05, Figure 4I). We also demonstrated that *CXCL9* increased in the serum of patients with lung cancer (n = 5, 1491 ± 663 pg/mL) compared to that in healthy volunteers (n = 605 ± 381 pg/mL) (*p* = 0.083, Figure 4J).

### 3.5. CD8^+^ T Cells Accumulated in A549-Induced Tumor Tissues and Suppressed Tumor Growth after Co-Injection with A549 Cells

To track the homing capacity of CD8^+^ T cells to the tumor microenvironment in the tumor xenograft mice in vivo, nivolumab, an anti-PD-1 antibody, was labeled with nuclear isotope indium-111 (In-111). There were four groups designed in this study, including A549 (Tumor 1, T1), A549 with PBMCs subcutaneously injected (Tumor 2, T2), A549 with PBMCs + A549 mixture (Tumor 3, T3), and A549 with PBMCs + PC9 mixture (Tumor 4, T4) (n = 3 for each group, Figure 5A). We found that there were significant radioactive signals in the T2, T3, and T4 tumors compared to the control, T1 (Figure 5A), revealing that CD8^+^ T cells have the migratory capacity for homing to tumor cells no matter whether co-injected with A549 or not (Figure 5B). Moreover, we demonstrated that only PBMCs co-injected with A549 cells presented high tumor suppression (T3) compared to the control group (T1) (Figure 5C).

### 3.6. Activation of Healthy PBMCs Inhibited Tumor Growth and Suppressed Radiotherapy-Derived Resistant Tumor Growth in the A549-Induced Tumor Xenografts In Vivo

Since we found that irradiation-treated A549 medium increased CD8^+^ T cell migration and activity, we intended to investigate whether radiotherapy affects CD8^+^ T cell cytotoxicity in an A549-induced tumor xenograft model in vivo. The in vivo experiment design was shown in Figure 6A, whereas A549 or A549 pre-exposed by irradiation was co-injected with PBMCs subcutaneously and parallel A549- (Tumor 1 (T1) and T3 in Figure 6A) and irradiated A549-induced tumor growth (T2 and T4 in Figure 6A) was measured. We demonstrated that healthy PBMCs significantly inhibited not only the co-injected A549 tumor growth but also the parallel tumor growth (T1 and T2 in Figure 6A,B left, *p* < 0.05). Although irradiation significantly suppressed the tumor volume (Figure 6A,B left), the tumor grew slowly at day 21 (Figure 6B right). We found that healthy PBMCs co-injection in the tumor xenografts inhibited the radioresistant tumor relapse (*p* < 0.05, Figure 6B right). As we know, CD8^+^ T cells exhausted in patients with lung cancer, the PBMCs from the patients with cancer were investigated in terms of the anti-tumor effect in the tumor xenografts (n = 3, P7, P9, and P10 in Table 1, Figure 6C). We used HCC827-derived tumors since HCC827-derived tumors in the xenograft mice were also suppressed by healthy PBMCs as demonstrated previously [21]. The results indicated that there was no significant tumor suppression in the patient’s PBMCs co-injected tumors and the parallel tumors (Figure 6C,D).

## 4. Discussion

Irradiation has been considered to damage cellular DNA to eradicate tumors as a promising tumor treatment in clinical practice. Besides the direct damage effect, the literature has revealed that irradiation can induce the cGAS-STING signaling pathway, mediating specific gene overexpression such as type I interferons [14]. The cGAS-STING signaling pathway is activated by broken DNA fragments. This self-protection mechanism is to defend from virus infection of the body. In virus infection, short viral DNAs or RNAs bind with cGAS and, consequently, with phosphorylates STING in ER lumen for activating IRF3 and NFkB transcriptional factors [25]. The process induces type I interferons overexpression and autocrine induces ISGs to suppress virus replication and secretions [14]. Based on this mechanism, radiotherapy is considered to not only directly damage tumor DNA but also induces type I interferons and other genes overexpression.

This study identified both *IFNα* and CD8^+^ T cell migratory chemokine *CXCL9* induced by 20 Gy irradiation in A549 cells, whereas *CXCL9* has been demonstrated to be a major CD8^+^ T cell migratory factor. *CXCL9* has been demonstrated to be regulated by NFkB and STAT1 transcriptional factors [26]; therefore, we speculated that irradiation-mediated expression of *CXCL9* may be induced by irradiation-activated NFkB and *IFNα*-activated STAT1 in A549 cells. An interesting finding in this study was that *CXCL9* also overexpressed in the healthy nonCD8^+^ PBMCs co-cultured with A549 (Figure 2E). To our knowledge, *CXCL9* is also released by activated CD103^+^ dendritic cells [24] and immunotherapy-treated macrophages [27]. The cellular mechanism that induces *CXCL9* expression is also dependent on the cGAS-STING signaling pathway-mediated type I interferons autocrine stimulation [28]. Therefore, we speculated that one of the mechanisms for the overexpression of *CXCL9* in nonCD8^+^ PBMCs was induced by either tumor- or activated dendritic cells-secreted type I interferons. The literature has indicated that immunotherapy targeting PD-1 and CTLA-4 specifically increases *CXCL9* and *CXCL10* from tumor-associated macrophages in tumor tissues, whereas *CXCL9* and *CXCL10* are induced by *IFNγ* in the tumor microenvironment [27]. Therefore, another possibility for increasing *CXCL9* in the nonCD8^+^ PBMCs was mediated by CD4^+^ T cells-secreted *IFNγ* since *IFNγ* is majorly secreted by CD4^+^ T cells, stimulated by antigen presenting cells [29].

In addition, *CXCL9* and *CXCL10* decreased in a patient’s nonCD8^+^ PBMCs (Figure 4I) but *CXCL9* concentration increased in the sera of patients with lung cancer (Figure 4J). The observation indicated that nonCD8^+^ PBMCs were continually stimulated by the tumor cells in the patients with lung cancer, resulting in the secretion of *CXCL9* in the serum. Hsin and colleagues have indicated that serum *CXCL9* is highly increased in patients with nasopharyngeal carcinoma which is associated with poor prognosis and survival rate [30]. We speculated that immune cells, such as dendritic cells and macrophages accumulating in tumor tissues responding to tumor cells for secreting in situ *CXCL9*, were vital for recruiting CD8^+^ T cells homing to tumor tissues. In contrast, serum *CXCL9* may not specifically guide CD8^+^ T cells homing to the tumor microenvironment.

CXCR3 and CXCR4 are the genes that respond to CD8^+^ T cell trafficking [23]; CXCR3 was selected for further investigation since we found that its ligand *CXCL9* was overexpressed in the irradiated A549 cells. We demonstrated that CXCR3 decreased in the CD8^+^ T cells of patients with lung cancer compared to those of healthy volunteers (Figure 4B), which has also been reported previously in patients with colorectal cancer [31]. Previous studies have demonstrated that CXCR3^−/−^ mice fail to facilitate CD8^+^ T cell migration without eliciting an anti-tumor immune response [32]. Moreover, the inhibition of CXCR3 reduces the immunotherapy-mediated migration and activation of CD8^+^ T cells [27]. Melvyn T. Chow and colleagues suggest that CXCR3 is required for the therapeutic efficacy of anti-PD-1 treatment [24]. Therefore, this implies that CXCR3 expression is critical for tumor patients as an immunotherapeutic prognosis marker. CXCR3 is suppressed by TGFβ in CD8^+^ T cells [33], whereas TGFβ is an immunosuppressive cytokine capable of inhibiting *GZMB*, *PRF1*, and *IFNγ* [34]. Activation of STAT3 is demonstrated to inhibit CXCR3 expression in CD8^+^ T cells [31,35]. Our data validated that CXCR3^low^PD-1^high^ CD8^+^ T cells exhibited poor migratory capacity toward tumors and failed to inhibit HCC827-derived tumor growth in vivo.

*GZMB*, *PRF1* and *IFNγ* are activation markers and PD-1 is the exhaustion marker of CD8^+^ T cells, whereas *GZMB* and *PRF1* are major hydrolytic enzymes leading to tumor apoptosis [36]. We found that parental A549 significantly induced CD8^+^ T cells’ expression of *GZMB* and *PRF1* (Figure 3A), which was derived by IL2 production (Figure 3B), leading to *GZMB* and *PRF1* overexpression in CD8^+^ T cells (Figure 3C). MHCI in tumor cells is recognized by the CD8 molecule to help stimulate CD8^+^ T cells; we speculated that this increases IL2 expression in the A549 co-cultured CD8^+^ T cells. Therefore, the irradiation-treated A549 medium exhibited no effect on *GZMB* and *PRF1* expression but rather increased *IFNγ* in CD8+ T cells (Figure 1F) since the medium contained *IFNα* which dramatically increased *IFNγ* (Figure 1G). However, the stimulation of CD8^+^ T cells was insufficient to suppress tumor growth with even CD8^+^ T cells still homing to tumors (Figure 5). We demonstrated that co-injected A549 with healthy PBMCs in the A549-derived tumor xenograft mice significantly suppressed tumor growth (Figure 6A,B). MHCI-presented neoantigens play a critical role in CD8^+^ T cell cytotoxicity that is mediated by antigen-presented cells, such as dendritic cells and macrophages, to stimulate the differentiation of naïve CD8^+^ T cells in lymph nodes. Our data support a cell therapy strategy for eradicating tumors [37]; a patient’s tumor-presented MHCI-neoantigens can be co-cultured with isolated PBMCs in vitro to activate CD8^+^ T cells. The activated CD8^+^ T cells, recognizing the patient’s tumors, can be injected back to the patient for eradicating tumors as a potential anti-tumor strategy in the future.

## 5. Conclusions

In this study, we found that irradiation-induced *IFNα* and *CXCL9* in lung cancer A549 cells result in the increase of CD8^+^ T cell motility homing to tumors (Figure 7). Previous studies have demonstrated that irradiation damages DNA and results in type I interferons expression. We identified that *IFNα* and CXCR9 were increased in irradiation-treated lung A549 cancer cells. *IFNα* specifically stimulated *IFNγ* expression in CD8^+^ T cells and synergistically enhanced *CXCL9* for CD8^+^ T cell migration. Meanwhile, we also demonstrated that decreased CXCR3, a receptor for *CXCL9*/10, in the CD8^+^ T cells of patients with lung cancers exhibited lower motility compared to healthy CD8^+^ T cells. This study revealed that healthy CD8^+^ T cells expressed hydrolytic *GZMB* and *PRF1* when co-cultured with A549 and significantly suppressed A549-induced tumor growth in vivo. Overall, we discovered a potential radiotherapy-augmented immunosurveillance mechanism to enhance and recruit CD8^+^ T cell homing to the tumor microenvironment.

## Figures and Tables

**Figure 1 biomedicines-09-01349-f001:**
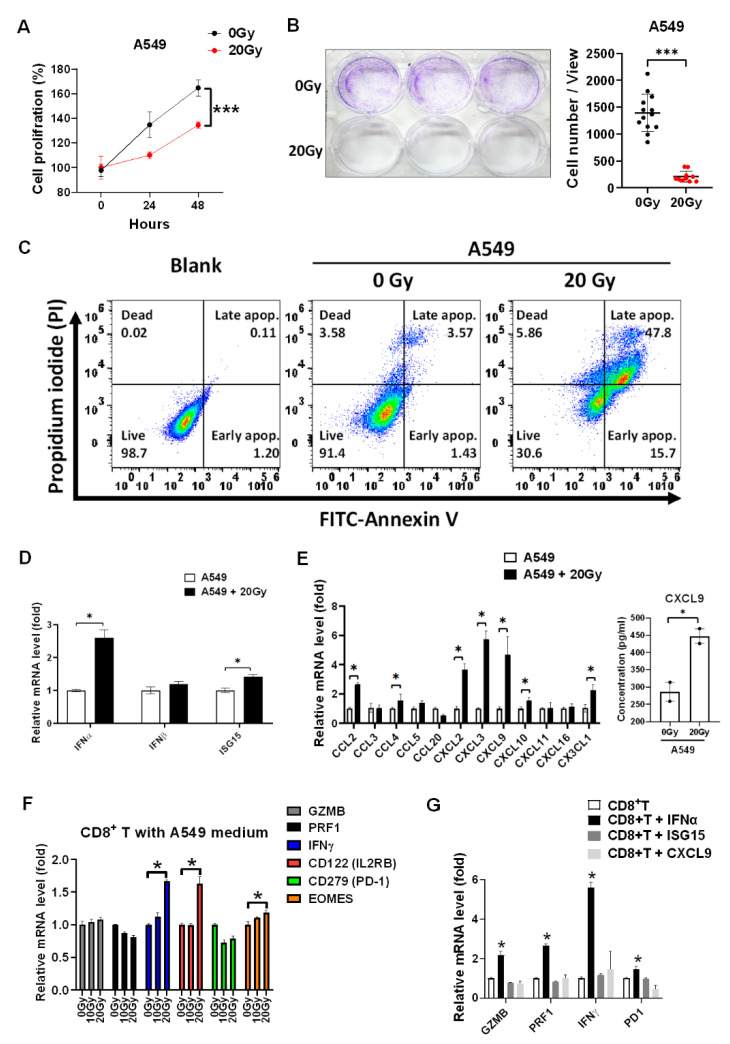
Irradiation inhibits A549 tumor growth and induces *IFNα* and chemokines overexpression to increase CD8^+^ T cell activity. (**A**) Cell viability was detected in A549 treated with 20 Gy of irradiation in a time-dependent manner. The quantification was calculated based on 3-times repeat. (**B**) Colony formation with 3-times repeat was measured in A549 treated with irradiation 7 days later. (**C**) Annexin V-labeled FITC was used to measure cell apoptosis in A549 treated with irradiation 24 h later using flow cytometry. (**D**) The type I interferons including α and β subtype and radiotherapeutic marker ISG15 were detected using qPCR in A549 treated with 20 Gy of irradiation 24 h later. (**E**) Meanwhile, the chemokines associating with CD8^+^ T lymphocyte trafficking were detected in A549 treated with irradiation 24 h later using qPCR. In addition, *CXCL9* concentration was measured using an ELISA technique in the cultured medium of irradiated A549 cells compared to parental A549 cells. (**F**) The A549 cultured medium after irradiation treatment was collected and incubated with PBMCs for 24 h and CD8^+^ T cells were consequently isolated for further investigation for the cellular levels of cytotoxic marker *GZMB*, *PRF1*, activation marker *IFNγ*, *CD122*, *EOMES*, and inhibitory marker *CD279* (PD-1) using qPCR. (**G**) *IFNα*, ISG15, and *CXCL9* (20 ng/mL) were selected and incubated with isolated CD8^+^ T cells for investigated whether which affected CD8^+^ T cell cytotoxicity (*GZMB* and *PRF1*), activity (*IFNγ*), and exhaustion (*CD279* (PD-1)). The quantification was calculated based on 3-times repeat in qPCR assay. * *p* < 0.05, *** *p* < 0.001.

**Figure 2 biomedicines-09-01349-f002:**
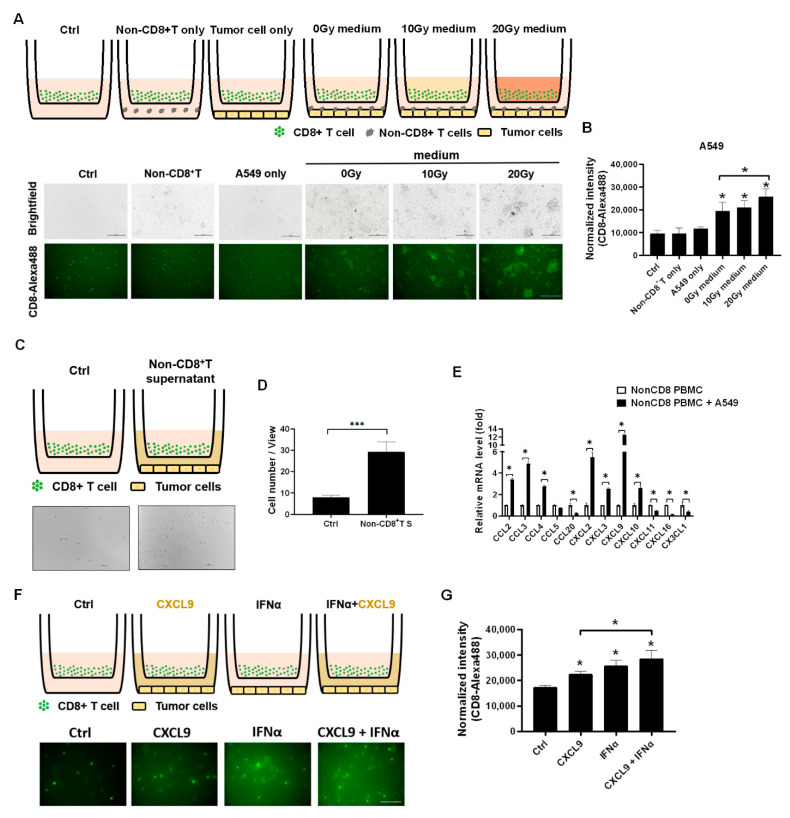
Irradiation-treated A549 medium and nonCD8^+^ PBMCs supernatant increased CD8^+^ T cell migration in vitro. (**A**) The isolated CD8^+^ T cells treated with the irradiation-treated A549 medium loaded on top of a 0.3 μm plate and lung cancer A549 or PC9 cells with nonCD8^+^ PBMCs loaded on down well for 24-h incubation. The CD8^+^ T cells were pre-labeled with Alexa488-labeled anti-CD8 antibody. (**B**) Quantitation of Alexa488 fluorescence representing CD8^+^ T cell migration activity was measured and compared based on 3-times repeat. (**C**) Since the A549 with nonCD8+ PBMCs enhanced CD8^+^ T cell migration, the supernatant of A549 with nonCD8^+^ PBMCs post 24 h was collected and investigated whether which increased CD8^+^ T cell migration. (**D**) CD8^+^ T cells migrating through 0.3 μm transwell were calculated and compared based on 3-times repeat. (**E**) The chemokines associated with CD8^+^ T cell trafficking were detected in nonCD8^+^ PBMCs after incubation with A549 for 24 h, including *CCL2*, *CCL3*, *CCL4*, *CCL5*, *CCL20*, *CXCL2*, *CXCL3*, *CXCL9*, *CXCL10*, *CXCL11*, *CXCL16*, and *CX3CL1*. (**F**) The healthy CD8^+^ T cells were treated with individual 20 ng/mL of *CXCL9* or *IFNα* for 24 h and (**G**) the Alexa488 fluorescence was captured and measured based on 3-times repeat. The quantification was calculated based on 3-times repeat in qPCR assay. Scale bar, 100 μm. * *p* < 0.05, *** *p* < 0.001.

**Figure 3 biomedicines-09-01349-f003:**
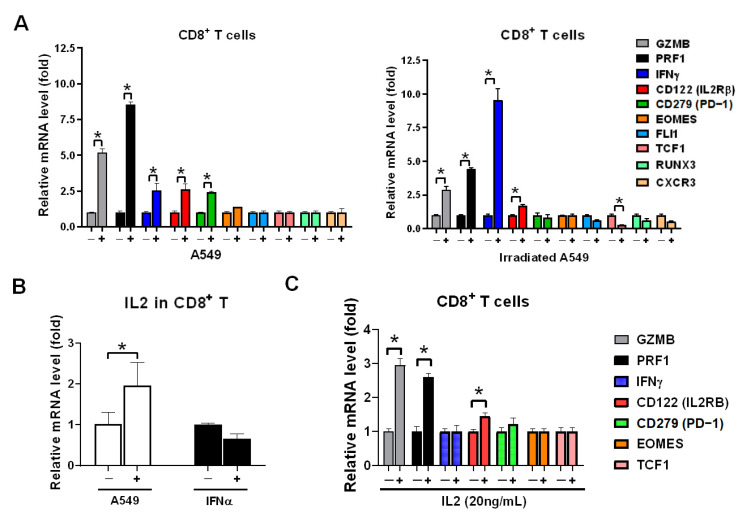
A549 cell lysate induced IL2 expression to stimulate *GZMB* and *PRF1* overexpression in CD8^+^ T cells. (**A**) Healthy CD8^+^ T cells were incubated with parental and irradiated A549 cells for 24 h, and which were isolated consequently and analyzed for expression of CD8^+^ T cells markers using qPCR, including the genes associated with the activation: *GZMB*, *PRF1*, *IFNγ*, *CD122* (IL2RB), inhibition: *CD279* (PD-1), transcriptional factor: *EOMES*, *FLI1*, *TCF1*, *RUNX3*, and migration receptor: CXCR3. (**B**) CD8^+^ T cells were individually treated with A549 cell lysate or 20ng/mL of *IFNα* for 2 h, and *IL2* expression was detected using qPCR. (**C**) CD8^+^ T cells were treated with 20 ng/mL of IL2 for 2 h, and the genes of CD8^+^ T cell markers were detected using qPCR. The quantification was calculated based on 3-times repeat in qPCR assay. * *p* < 0.05.

**Figure 4 biomedicines-09-01349-f004:**
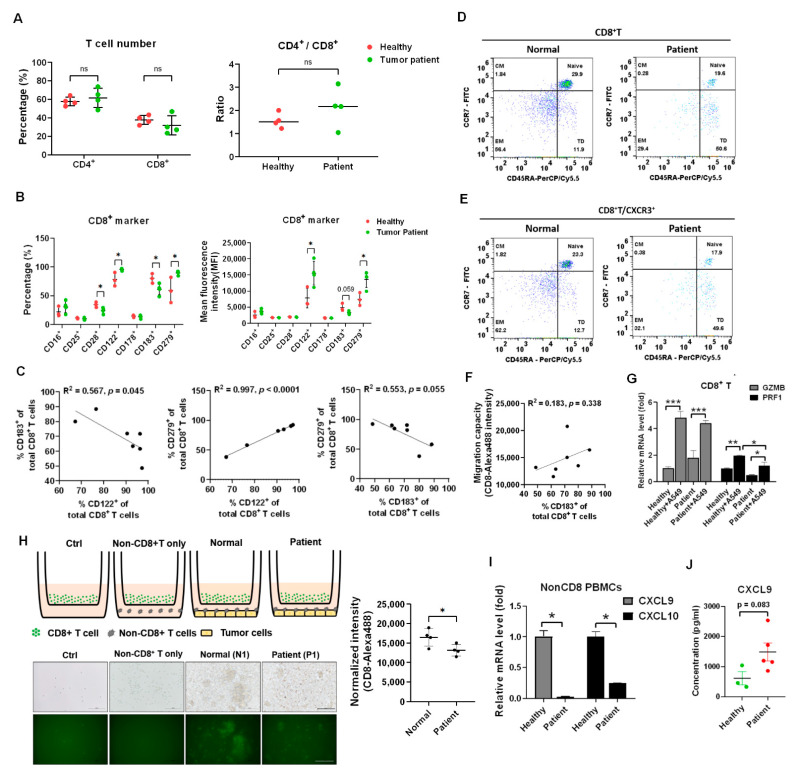
Reduction of CD8^+^ T cells migration in vitro in the patients with lung cancer. (**A**) The cell percentage of CD4^+^ T and CD8^+^ T cells was measured between the healthy volunteers and the patients with lung cancer. The CD4^+^/CD8^+^ ratio was also compared. ns: non-significance. (**B**) Th CD8^+^ T cell markers were detected and compared between the healthy volunteers and the patient with lung cancer, including activation markers: CD16, CD25, CD28, *CD122*, CD178, migratory marker: CD183 (CXCR3), and inhibitory marker: *CD279* (PD-1). (**C**) The correlation among *CD122*, CD183, and *CD279* was analyzed. (**D**) CD8^+^ T cells between the healthy volunteers and the patients with lung cancer were classified into four subsets based on the expression of CD45RA and CCR7 by flow cytometry (Naïve: CD45RA^+^CCR7^+^; Central-memory (CM): CD45RA^−^CCR7^+^; Effector-memory (EM): CD45RA^−^CCR7^−^; and Terminally differentiated (TD): CD45RA^+^CCR7^−^). (**E**) The CD183 (CXCR3) expression was detected in the four subsets of CD8^+^ T cells. (**F**) The correlation between CD183 (CXCR3) expression and the CD8^+^ T cell migration ability was analyzed in the healthy volunteers and the patients with lung cancer. (**G**) The A549 cells with incubated with PBMCs from a healthy volunteer and a patient with lung cancer for 24 h, and CD8^+^ T cells were isolated and analyzed for *GZMB* and *PRF1* expression by qPCR. (**H**) The CD8^+^ T cells migration was compared between the healthy volunteers (n = 4) and the patients with lung cancer (n = 4). Scale bar, 100 μm. (**I**) The non-CD8^+^ PBMCs between a healthy volunteer and a patient with lung cancer were analyzed for *CXCL9* and *CXCL10* expression. (**J**) The *CXCL9* concentration was measured using an ELISA technique between the healthy volunteers (n = 3) and the patients with lung cancer (n = 5). The quantification was calculated based on 3-times repeat in qPCR assay. * *p* < 0.05, ** *p* < 0.01, *** *p* < 0.001.

**Figure 5 biomedicines-09-01349-f005:**
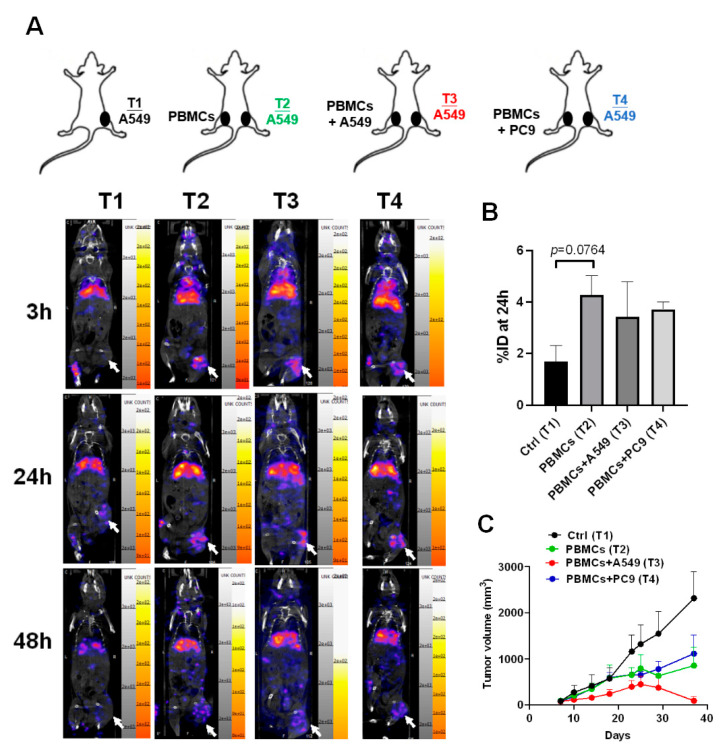
CD8^+^ T cells migrated to the tumor microenvironment and significantly suppressed tumor growth in the A549-derived tumor xenografts co-injected with healthy PBMCs and A549 cells. (**A**) A549 cells (2 × 10^6^) were injected subcutaneously into the right legs of mice (Tumor 1 (T1) to T4), and after the tumor grew up 100 mm^3^, the PBMCs (1 × 10^7^) alone or with A549 or PC9 (2 × 10^5^) were injected into the left for 48 h circulation. ^111^In-labeled nivolumab was injected into the mice and the radioactive signals were measured in 3, 24, and 48 h. Tumors are indicated by arrows. (**B**) Quantification for the radioactive signals of 24 h in the T1 to T4 tumors was measured based on an 8μCi calibrated standard. (**C**) Meanwhile, the tumor volumes in T1 to T4 were recorded and compared. Tumors are indicated by arrows. n = 3 for each group.

**Figure 6 biomedicines-09-01349-f006:**
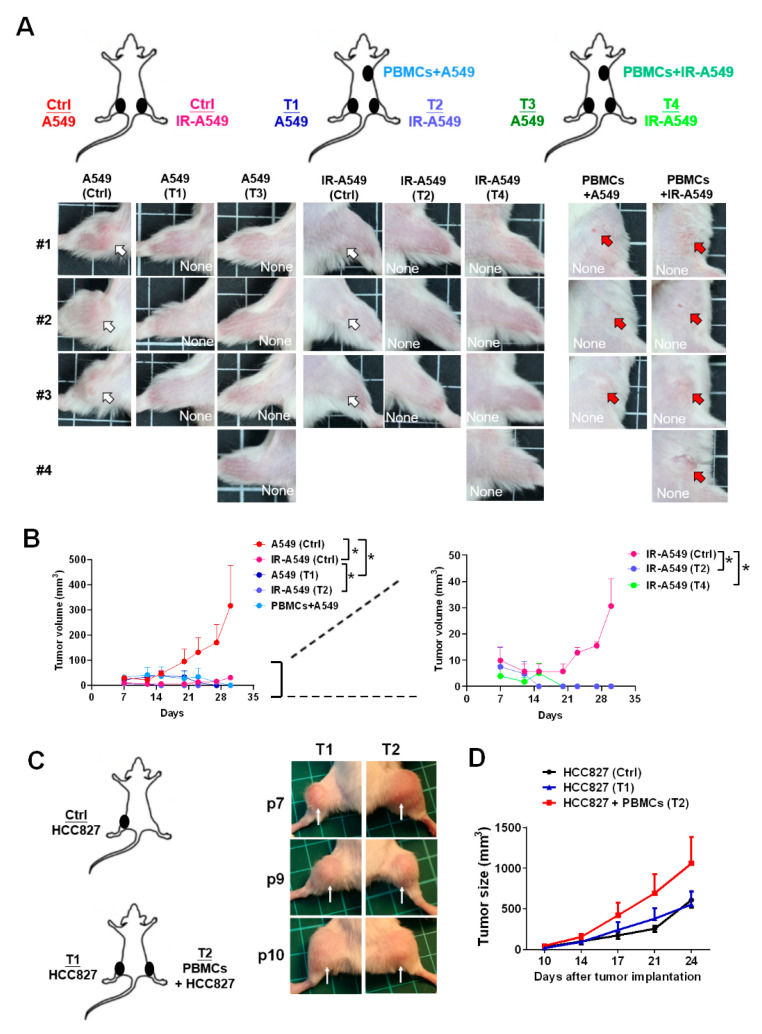
Healthy PBMCs con-injected with tumor cells significantly inhibited A549 tumor growth. (**A**) The A549 or irradiation (IR)-treated A549 cells were injected subcutaneously in a mouse to establish the tumor xenografts. Meanwhile, PBMCs with A549 or irradiated A549 were co-injected subcutaneously on the back. The tumor images at day 30 were captured and shown. (**B**) The tumor burdens were measured and compared consequently by day 7, 12, 15, 20, 23, 27, and 33, whereas control mice: n = 3, PBMCs + A549: n = 3, PBMCs + IR-A549: n = 4. (**C**) The anti-tumor effect of PBMCs from the patients with lung cancer (n = 3, P7, P9, and P10 in Table 1) was investigated in an HCC827-derived tumor xenograft model. The control group: n = 3. The tumors at day 24 for T1 and T3 were shown, (**D**) and the tumor sizes were recorded on day 10, 14, 17, 21, and 24. Tumors are indicated by arrows. Tumor volumes were calculated according to the formula: 0.52 × width^2^ × length, herein the width represents the smaller tumor diameter. * *p* < 0.05.

**Figure 7 biomedicines-09-01349-f007:**
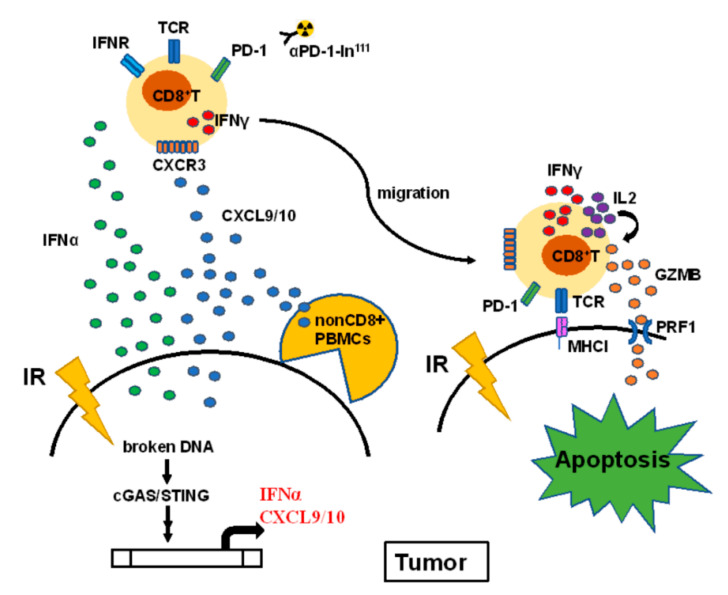
The proposed hypothesis illustrates the potential mechanism of radiotherapy augmenting CD8^+^ T cells against NSCLC. Irradiation (IR) activates the cGAS-STING signaling pathway through the broken DNA fragments, resulting in *IFNα* and *CXCL9* overexpression and secretion. The secreted cytokines increase *IFNγ* expression and recruit CD8^+^ T cells through CXCR3 receptor homing to the tumor microenvironment. The in^111^ labeled PD-1-antibody was used to detect CD8^+^ T cells homing to tumors. CD8^+^ T cells contact A549 cells to elicit autocrine IL2-mediated *GZMB* and *PRF1* expression, resulting in tumor apoptosis. Particularly, PBMCs co-injected with A549 significantly suppressed tumor growth in the A549-derived tumor xenografts in vivo.

**Table 1 biomedicines-09-01349-t001:** Information on the enrolled volunteers.

Sample Number	Age	Gender	Cancer Type	Treatment
N1	42	Male	-	-
N2	40	Male	-	-
N3	39	Male	-	-
N4	27	Male	-	-
P1	68	Female	Lung cancer (late stage)	None ^a^
P2	71	Male	Lung cancer (late stage)	None ^a^
P3	74	Female	Lung cancer (late stage)	None ^a^
P4	51	Female	Lung cancer (late stage)	None ^a^
P7	75	male	Hard palate	Nivolumab ^b^
P9	54	female	Skin	Nivolumab ^b^
P10	52	male	Hypopharynx	Pembrolizumab ^b^

^a^ The blood samples were collected before clinical therapies. ^b^ The blood samples were collected after immunotherapies.

**Table 2 biomedicines-09-01349-t002:** The primer sequence for qPCR.

Gene	Direction	Primer Sequence
*GAPDH*	Forward	GAGTCAACGGATTTGGTCGT
	Reverse	TTGATTTTGGAGGGATCTCG
*IFNA*	Forward	GCCATCTCTGTCCTCCATGA
	Reverse	ATCTCATGATTTCTGCTCTGACAA
*IFNG*	Forward	TCCCATGGGTTGTGTGTTTA
	Reverse	AAGCACCAGGCATGAAATCT
*ISG15*	Forward	TGTCGGTGTCAGAGCTGAAG
	Reverse	GCCCTTGTTATTCCTCACCA
*CCL2*	Forward	CCCCAGTCACCTGCTGTTAT
	Reverse	TGGAATCCTGAACCCACTTC
*CCL3*	Forward	TGCAACCAGTTCTCTGCATC
	Reverse	TTTCTGGACCCACTCCTCAC
*CCL4*	Forward	AAGCTCTGCGTGACTGTCCT
	Reverse	GCTTGCTTCTTTTGGTTTGG
*CCL5*	Forward	CGCTGTCATCCTCATTGCTA
	Reverse	GAGCACTTGCCACTGGTGTA
*CCL20*	Forward	GCAAGCAACTTTGACTGCTG
	Reverse	ATTTGCGCACACAGACAACT
*CXCL2*	Forward	GCAGGGAATTCACCTCAAGA
	Reverse	GGATTTGCCATTTTTCAGCA
*CXCL3*	Forward	GCAGGGAATTCACCTCAAGA
	Reverse	GGTGCTCCCCTTGTTCAGTA
*CXCL9*	Forward	TTTTCCTCTTGGGCATCATC
	Reverse	TCAATTTTCTCGCAGGAAGG
*CXCL10*	Forward	CTGTACGCTGTACCTGCATCA
	Reverse	TTCTTGATGGCCTTCGATTC
*CXCL11*	Forward	AGAGGACGCTGTCTTTGCAT
	Reverse	TAAGCCTTGCTTGCTTCGAT
*CXCL16*	Forward	ACTCGTCCCAATGAAACCAC
	Reverse	ATGAAGATGATGGCCAGGAG
*CX3CL1*	Forward	GACCCCTAAGGCTGAGGAAC
	Reverse	CTCTCCTGCCATCTTTCGAG
*GZMB*	Forward	ACTGCAGCTGGAGAGAAAGG
	Reverse	TTCGCACTTTCGATCTTCCT
*PRF1*	Forward	ACTCACAGGCAGCCAACTTT
	Reverse	GGGTGCCGTAGTTGGAGATA
*CD122* (IL2RB)	Forward	GCTGATCAACTGCAGGAACA
	Reverse	TGTCCCTCTCCAGCACTTCT
*CD279* (PD-1)	Forward	GTGTCACACAACTGCCCAAC
	Reverse	CTGCCCTTCTCTCTGTCACC
*EOMES*	Forward	CCACTGCCCACTACAATGTG
	Reverse	TTCCCGAATGAAATCTCCTG
*TCF1*	Forward	AGCCAAGGTCATTGCAGAGT
	Reverse	GTGGTGGATTCTTGGTGCTT
*RUNX3*	Forward	CAGAAGCTGGAGGACCAGAC
	Reverse	TCGGAGAATGGGTTCAGTTC
*CD183* (CXCR3)	Forward	ACACCTTCCTGCTCCACCTA
	Reverse	GTTCAGGTAGCGGTCAAAGC

## Data Availability

Not applicable.

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
