# Peer review of "Irradiation Mediates IFNα and CXCL9 Expression in Non-Small Cell Lung Cancer to Stimulate CD8+ T Cells Activity and Migration toward Tumors"

_biomedicines, 2021, doi:10.3390/biomedicines9101349_

Round 1

Reviewer 1 Report

The manuscript submitted by Cheng et al. reports the potential radiotherapy-augmented immunosurveillance mechanism to enhance and recruit CD8+ T cell homing to the tumor microenvironment.

It is well written, the methodology appropriate including a variety of techniques and the conclusions are original and consistent with the results.

Listed below are some suggested changes:

Point 1: In the Material and methods section, too much technical information is presented. Maybe a good idea would be to keep some essential information and to put the rest of it in the supplementary material (as for tables 1 and 2).

Point 2: In figure 1 A, for the graph concerning the cell viability assay, the indication of the viability percentage is not shown on the axis. Insert it.

Point 3: How many times have the experiments been conducted with each single method? It is not reported in figure legends and it is not clear in the text. It is important to report it to understand the "statistically significant" that you show. Review materials and methods and figure legends by entering missing information.

Author Response

  1. We appreciate your comment. Some detail for the common techniques is moved and shown in Table S2.
  2. We appreciate the comment and apology the missing label. The Y-axis Cell viability (%) is inserted in Fig 1A.
  3. We appreciate the constructive comment. We have added the repeat number in the Methods and the Figure legends.

Reviewer 2 Report

This is an excellent translational article, with several experiments suggesting a directly triggering of irradiation of immuno-mediated mechansims underlying tumor suppression. The article is well performed and seems to have all ethical permissions. In order to improve the article, I would suggest to test CXCL9 chemokine levels also in serum of patients with non-small-cell lung cancer (NSCLC) irradiated vs. non irradiated. This point might further support a systemic activation of inflammation induced by irradiation in this oncological setting. I believe that a pilot study with a small numebr of patients might be sufficient to explore this point.

Author Response

We appreciate your constructive comment. We have added CXCL9 concentration in the medium of irradiated A549 (Fig 1E) and in the serum of patients with lung cancer (Fig 4J). CXCL9 is measured using ELISA, that is also added and described in the Methods. For the observation, we have added a paragraph in Discussion “In addition, CXCL9 and CXCL10 decreased in a patient’s nonCD8+ PBMCs (Fig 4I) but CXCL9 concentration increased in the sera of patients with lung cancer (Fig 4J). The observation indicated that nonCD8+ PBMCs were continually stimulated by the tumor cells in the patients with lung cancer, resulting in secretion of CXCL9 in the serum. Hsin and colleagues have indicated that serum CXCL9 highly increases in patients with nasopharyngeal carcinoma associating with poor prognosis and survival rate [37]. We speculated that immune cells such as dendritic cells and macrophages accumulated in tumor tissues responding to tumor cells for secreting in situ CXCL9 was vital to recruit CD8+ T cells homing to tumor tissues. In contrast, serum CXCL9 may not specifically guide CD8+ T cells homing to tumor microenvironment.”

Round 2

Reviewer 2 Report

no further comments